

# Tsunami arrival time detection system applicable to discontinuous time-series data with outliers

Jun-Whan Lee [1], Sun-Cheon Park[1], Duk Kee Lee[1], Jong Ho Lee[1]

[1]National Institute of Meteorological Sciences, 33, Seohobuk-ro, Seogwipo-si, Jeju-do, 63568, Rep. of Korea

*Correspondence to*: Sun-Cheon Park (suncheon@korea.kr)

**Abstract.** Timely detection of tsunamis with water-level records is a critical but logistically challenging task because of outliers and gaps. We propose a tsunami arrival time detection system (TADS) that can be applied to discontinuous time-series data with outliers. TADS consists of three major algorithms that are designed to update at every new data acquisition: outlier detection, gap-filling, and tsunami detection. To detect a tsunami from a record containing outliers and gaps, we

propose the concept of the event period. In this study, we applied this concept in our test of the TADS at the Ulleung-do surge gauge located in the East Sea. We calibrated the thresholds to identify tsunami arrivals based on the 2011 Tohoku tsunami, and the results show that the overall performance of TADS is effective at detecting a small tsunami signal superimposed on both an outlier and gap.

## 1 Introduction

A tsunami is one of the most devastating natural phenomena caused by several events such as seaquakes, submarine landslides, terrestrial landslides, volcanic eruptions, asteroid and comet impacts, and man-made explosions (Pugh and Woodworth, 2014). The eastern coast of the Korean Peninsula is not exempt from tsunamis because of the high tsunami energy concentration at the coast based on the peculiar topographic conditions of the East Sea where the Yamato rise strongly affects the propagation of tsunamis (Cho and Lee, 2013). A low probability exists for tsunamis to occur in the East

Sea. However, if and when they do occur, they pose a high risk of damage to not only Korea but also neighboring countries. For example, two tsunamis in 1983 and 1993, which originated near the western coast of Akita and Hokkaido, Japan, respectively, caused severe damage along the eastern coast of the Korean Peninsula.

The Korea Meteorological Administration (KMA), as the government's meteorological organ, is responsible for issuing information on tsunamis. A tsunami warning based only on the occurrence of an earthquake could lead to false alarms. Thus,

to monitor tsunamis, the KMA has operated a surge gauge (aerial ultrasonic type) at Ulleung-do since 1999. The Ulleung-do, located in the East Sea, plays a critical role in tsunami hazard mitigation of the Korean Peninsula because it can confirm the approach of tsunamis 30 min or more before they impact the eastern coast (Fig. 1). However, when the Tohoku, Japan tsunami occurred in 2011, even though a post analysis revealed it to be a small tsunami (less than 0.3 m), the KMA could not announce important properties of the tsunami such as its arrival time and wave height in a press release because of lack of a




tsunami detection system (Yoon et al., 2012). Thus, the development of a tsunami detection system that automatically provides prompt notification is necessary.

Detecting tsunamis using the Ulleung-do surge gauge is logistically challenging because onshore measurements are usually associated with high background noise (Joseph, 2011). In addition, outliers derive from various problems related, for

example, to meteorological events or electrical malfunctions occurring in water-level sensor data streams. Because outlier detection (as well as the anomaly detection, despiking, and noise removal) has been researched for a long time, various techniques have been developed. Ehrentreich and Sümmchen (2001) used a wavelet transform method to remove the spikes from the Raman spectra. Feuerstein et al. (2009) developed a despiking algorithm based on filtering methods using clinical data. Goring and Nikora (2002) and Jesson et al. (2013) presented a phase-space thresholding method that is applied to

automated post-processing software to remove spikes from acoustic Doppler velocimeter data (Jesson et al., 2015). However, these methods require a complete set of data or data in a batch. Thus, they are not suitable for real-time or near real-time applications (Hill et al., 2009). To perform in real time or near real-time, the outlier detection algorithm must consider the data stream sequentially or the outlier should be detected immediately after it appears. Several studies have defined a window that steps through the data stream to operate in real time. The most up-to-date survey on the window-based outlier

detection algorithm was provided by Gupta et al. (2014). Yamanishi and Takeuchi (2002) developed an on-line discounting learning algorithm that gradually forgets the effect of past data. Hill et al. (2009) developed an outlier detection algorithm based on dynamic Bayesian networks that adds new state variables over time. Hill and Minsker (2010) developed an outlier detection algorithm based on a data-driven univariate autoregressive model and corresponding prediction interval. However, because most of these algorithms predict the subsequent set of chronologically sequential data using soft computing

techniques, which are advantageous in detecting a short-term outlier, these applications are not suitable for long-term outliers. They also require huge memory and considerable computation time.

Moreover, water-level sensors often experience unexpected gaps or missing points that cause major difficulties in detecting tsunamis. These difficulties are explained by such occurrences as: a failure of the recording or interruption of the communication network, aging equipment, and mistakes by field staff (Ustoorikar and Deo, 2008). When data are lost or

25 incomplete, the tsunami detection algorithms that require several hours of data get stopped even after the recording is restarted. For short gaps, a linear interpolation may be used. Otherwise, for long gaps that include irregular patterns, a different algorithm is required. Several kinds of soft computing techniques for long gaps have been developed. These include chaos theory, genetic programming, empirical orthogonal functions, and artificial neural networks (ANNs) (Elshorbagy et al., 2002; Nitsure et al., 2014; Tolkova, 2009; Pashova and Popova, 2011). Recently, Lee and Park (2016) developed a gap-

30 filling algorithm based on ANNs and an end-point fixing method (EPFM) and applied it to hourly data from a tide station. As previously mentioned concerning the outlier detection algorithm, although soft computing techniques are more accurate, these applications require considerable computing time. Thus, Lee and Park (2015) modified Lee and Park (2016)'s method and used a moving average filter rather than an ANNs. The algorithm was successfully tested to the records of the Ulleung-do tide station.



Because of the infrequent occurrence of large tsunamis, an important technical requirement for detecting tsunamis is a sensitive tsunami detection algorithm. This can be used to detect weak tsunami signals that are common in the Ulleung-do surge gauge. To detect a tsunami, several tsunami detection algorithms have been developed based on specific purposes and limitations. Mofjeld (1997) developed a deep-ocean assessment and reporting of tsunamis (DART) algorithm that uses a cubic polynomial fit to the data over the preceding three hours (Meinig et al., 2005). Beltrami (2008) modified the DART algorithm based on the ANNs to update the coefficients of every sampling interval. Beltrami and Risio (2011) proposed a tsunami detection algorithm that combines the DART algorithm with an infinite impulse response-time domain digital filter to filter out any disturbances derived from wind waves. Because the DART algorithm does not provide information on tsunami height but only arrival time, Beltrami (2011) extended the length of the interval between the actual and prediction times. Bressan and Tinti (2012) proposed a tsunami early detection algorithm (TEDA) designed to detect an anomalous water level based on two slope-based algorithms: tsunami detection and secure detection. The TEDA was successfully calibrated and tested on both synthetic tsunamis and historical tsunami records (Bressan and Tinti, 2012; Bressan et al., 2013).

Overall, a tsunami detection system should be designed for the detection of a small tsunami signal superimposed on both an outlier and gap. This study presents a tsunami detection system applicable to discontinuous time-series data with outliers, which we call tsunami arrival time detection system (TADS). The 10-second interval data of the Ulleung-do surge gauge recorded from March 1–31, 2011 were employed to demonstrate the performance of the TADS, in which not only outliers and gaps but also the 2011 Tohoku tsunami signals were included (Fig. 2). Outliers that may have resulted from meteorological events were found in similar periods identified in special weather reports. In addition, suspicious gaps lasting 6 h were found in the data of the day before the 2011 Tohoku earthquake.

## 2 Methods

The flow of the proposed TADS is shown in Fig. 3 and is divided into three major algorithms designed to update at every new data acquisition. These algorithms concern: outlier detection, gap filling, and tsunami detection. First, for the purpose of TADS, the outlier detection algorithm should accommodate the characteristics of data streams of water-level sensors, that is, gaps and long-term outliers. Therefore, we developed an outlier detection algorithm viable for discontinuous time-series data with long-term outliers (Lee et al., 2016). Second, because the gap-filling algorithm should manage gaps as soon as a new data point becomes available, we divided the gap-filling algorithm into two categories according to gap size. One-point and short gaps are replaced by nearest neighbor and linear interpolation, respectively. Hereafter, these methods are collectively referred to as the short gap-filling algorithm (SGFA) in a batch. For long gaps, we followed the method of Lee and Park (2015), which is hereafter referred to as long gap-filling algorithm (LGFA). Finally, to detect weak tsunami signals and reduce the probability of false alarms, three kinds of tsunami detection algorithms are applied to TADS. One is the DART algorithm (Mofjeld, 1997), which is an amplitude-based algorithm using Newton's formula for forward extrapolation.



Another is the tsunami detection module of TEDA (Bressan and Tinti, 2012), which is a slope-based algorithm and is hereafter referred to as a SLOPE algorithm. The other is newly developed and known as the TIDE algorithm, which is an amplitude-based algorithm employing harmonic analysis. To distinguish a tsunami from a record that may contain contributions from swells, local seiches, storm surges, and so on (Joseph, 2011), the tsunami detection algorithm is designed

to work only when an earthquake occurs, whereas the outlier detection and gap-filling algorithms terminate during the event period. In the remainder of this section, we describe each algorithm in detail.

## 2.1 Outlier detection algorithm

Fig. 4 presents a flow of the outlier detection algorithm. The basic concept of this algorithm as developed in this study is that the point at which the difference in wave height between neighboring points surpasses the threshold is designated as an

10 outlier. A delayed sliding window is introduced to detect an outlier without using any soft computing techniques. The sliding window is composed of 8-point data in which each point is numbered in ascending order from zero ($p_0$) to seven ($p_7$). Because the sampling interval of the Ulleung-do surge gauge data is 10 s, such a delay does not affect the overall efficiency of the TADS in generating a timely alarm. The wave height and its time information of point are hereafter referred to as h and t with a subscript indicating the point number. In addition, the target point $p_t$, which determines whether the point is an

15 outlier or not, is located at the 5th point from the present time $t_{now}$.

Unlike a single outlier, detecting long-term outliers that maintain a sudden change for a certain period is difficult. To deal with long-term outliers, the proposed outlier detection algorithm employs start mode (MODE = 0), keep mode (MODE = 1), and end mode (MODE = 2). In the start mode, the outlier detection algorithm searches for the point at which the outlier starts. Once the outlier is detected, the keep mode accelerates the algorithm. Then, the end mode searches for the point at which the

20 outlier ends and continues until satisfying one of the stopping conditions.

One common problem in outlier detection is the presence of gaps in the data stream. Considering a situation in which an outlier follows a long gap, a wave height $h_g$ and its timing $t_g$ of the last point before the gap are introduced. This is referred to as a keeping method in this study. If the difference between $h_g$ and $h_5$ exceeds a given threshold THh in a situation in which $p_5$ but not $p_6$ is measured, $p_t$ is designated as an outlier. Alternatively, $p_t$ is classified as an outlier depending on the

25 difference between $h_5$ and the wave heights of the other points from $p_0$ to $p_4$. TH0, TH1, TH2, TH3, and TH4 are the respective thresholds for cases in which the sign of the difference between $h_5$ and the wave heights of the other points from $p_0$ to $p_4$ is a positive number (plus sign) or zero. By contrast, TH_0, TH_1, TH_2, TH_3 and TH_4 are the thresholds for the cases in which the the sign of difference is a negative number (minus sign). In other words, we assign different thresholds depending not only on the distance between points but also the sign of the water-level change.

Once the outlier is detected by the thresholds, MODE changes from 0 (start mode) to 1 (keep mode) and TAG stores the number of intermediate points between $p_5$ and the other point compared. After designating the intermediate points as outliers,



MODE becomes 2. In addition, the sign S, wave height $h_s$ and its timing $t_s$ of the point at which the outlier starts are stored in memory for use in the end mode. The sign is based on a sign function of a real number x and can be expressed as:

$$sgn(x) = \begin{cases} -1 & if\ x < 0, \\ 0 & if\ x = 0, \\ 1 & if\ x > 0. \end{cases} \quad (1)$$

The end mode is designed to identify long-term outliers. The end mode ceases to designate $p_t$ as an outlier when the slope is
dramatically changed or the difference between $h_5$ and $h_s$ exceeds the threshold. THS1, THS2, THS3, and THS4 are the thresholds for the end mode that depend on the gap conditions in the window. Different conditions are required as time passes because of the tide that changes the mean water level. Thus, THD1 and THD2 are introduced to stop the end mode and depend on the time interval, even though the difference between $h_5$ and $h_s$ falls within the bounds of the threshold.

**2.2 Gap-filling algorithm**

Fig. 5 presents a flow of the gap-filling algorithm. Because the gap-filling algorithm is activated after the outlier detection algorithm, not only gaps of original data but also outliers detected by the outlier detection algorithm are subject to the gap-filling algorithm. The first step is to count the gap size $n_{Gap}$ because the gap-filling algorithm is divided in two (SGFA and LGFA) according to the gap size. The SGFA is applied if the gap is smaller than $n_{LGFA}$. If the gap size equals 1, then the gap is replaced by the wave height at time $t_{now} - 1$. Otherwise, if the gap size is greater than 1 but smaller than a predefined
length $n_{LGFA}$, then a linear interpolation is applied to fill the gap. A predefined length of points (N_inter) from the last point before and the first point after the gap is used to create the linear interpolation in order to prevent a distortion of slope caused by temporary water-level fluctuation.

The LGFA is applied if the gap size is greater than $n_{LGFA}$. The LGFA applies the hypothesis that the missing data will follow the trend of the past water-level movement. Thus, the data in the gap period should not include events of both long period
waves such as tsunamis and storm surges, and short period waves such as wind waves, which change the water level temporarily. To filter out background noise, a moving average filter that sets a time interval at $mv_{LGFA}$ is applied to the data stream. The first step of LGFA is to define the target window that consists of gaps, end points ($EP1$ and $EP2$), and target data. The size of the target window is predefined as $windowsize \times (n_{Gap} + 2)$, where $windowsize$ is a coefficient that determines the size of the window. In addition, the search window is set to the same size as the target window from the last
point before the gap. The search window then moves toward the last point of the past dataset $m_{search}$, calculating the mean absolute error (MAE) between the target and search data. The size of $m_{search}$ is $npastdata \times windowsize \times (n_{Gap} + 2)$ or $npastdata$ times the target window size. To follow the hypothesis, we selected the search window that shows the minimum MAE after deleting the MAEs that occur during the event period $t_{event}$. The SW data, which is the remaining data of the selected search window concealed in the search data, are rebalanced by the EPFM. With the first and last points of SW data
fixed to the $EP1$ and $EP2$, the intermediate data are linearly balanced using the following equation:



$$H_{EP}(t) = H_{ori}(t) + \frac{b-a}{d}(t-c) + a \quad for \ c \le t \le c+d \tag{2}$$

where $H_{EP}(t)$ is the water level after the EPFM is applied, $H_{ori}(t)$ is the water level before the EPFM is applied, $a$ is $h_1 - h_2$, $b$ is $h_4 - h_3$, $c$ is $t_1$, and $d$ is $t_2 - t_1$. Finally, the SWEP data, which are not only selected but also rebalanced, are used to replace the gaps. A more detailed description of LGFA can be found in Lee and Park (2015).

## 5 2.3 Tsunami detection algorithm

Fig. 6 presents a flow of the tsunami detection algorithm. When the time $t_{now}$ is under the event period $t_{event}$, three distinctive tsunami detection algorithms, denoted by DART, SLOPE, and TIDE, run simultaneously during every new data acquisition. The DART algorithm is an amplitude-based algorithm that uses a cubic polynomial fit to the data stored over the preceding 3 h and 10 min to predict the water level at $t_{now}$ and can be expressed as:

$$10 \quad h_{DART}(t_{now}) = \sum_{i=0}^{3} \omega_i \bar{h}_i \left( t_{now} - 1 - \frac{300}{\Delta t} - i \frac{3600}{\Delta t} \right) \tag{3}$$

where $h_{DART}$ is the predicted water level, $\bar{h}$ is the 10-minute average of the measured water level, $\Delta t$ is a sampling interval expressed in seconds, and $\omega_i$ are the coefficients calculated by applying Newton's forward divided difference formula. Those variables for the case of $\Delta t = 10\,s$ are listed in Table 1. If the DART index or absolute value of the difference between h($t_{now}$) and $h_{DART}$($t_{now}$) surpasses a given threshold $TH_{DART}$, then an alarm(DART) is triggered.

15 The SLOPE algorithm is a slope-based algorithm designed to detect a tsunami with an impulsive front. Using the same terminology as Bressan and Tinti (2011), we obtain the detided instantaneous signal $IS(t)$ by subtracting the estimated slope of the tide $Tide(t)$ from the instantaneous signal $IS_T(t)$. The background slope $BS(t)$ is calculated using one of the three methods proposed by Bressan and Tinti (2011) that was found to be the most efficient for the Ulleung-do surge gauge data, and can be expressed as

$$20 \quad BS(t) = \text{standard deviation of } IS(t') \cdot \sqrt{2}; \quad t' \in I_{BS} \tag{4}$$

where $I_{BS}$ is the time interval of $BS(t)$. The ratio of $|IS(t)|$ to $BS(t)$ is known as the control function $CF(t)$. The SLOPE algorithm triggers alarm(IS) and alarm(CF) every time the absolute values of IS($t_{now}$) and CF($t_{now}$) pass a given threshold $TH_{IS}$ and $TH_{CF}$, respectively. A more detailed description can be found in Bressan and Tinti (2011). Parameter configuration calibrated on the Ulleung-do surge gauge records are listed in Table 1. It should be noted that even though we used 10 s

25 interval data of Ulleng-do surge gauge in a manner different from that of Bressan and Tinti (2011) by calibrating with one-minute interval data, the same values of all parameters except thresholds yield the best performance.

The TIDE algorithm is an amplitude-based algorithm based on harmonic analysis. When the time $t_{now}$-1 does not belong to the event period $t_{event}$, T_TIDE is activated to predict the tide data $h_{Tide}(t)$ over the period $t_{FP}$. T_TIDE is a classical harmonic analysis program that evaluates the tidal constituents (frequency, amplitude, phase, etc.). A detailed description can be found

30 in Pawlowicz et al. (2002). To increase the speed of T_TIDE, we applied the sampled data $h_s(t)$ collected from the water-level data h(t) regularly spaced at an sampling interval $t_{sample}$ over the preceding time $t_{BP}$. Once $h_{Tide}(t)$ is calculated, the TIDE algorithm skips the aforementioned process for the time interval of duration $t_{FP}$. The detided water-level data $h_{Detide}(t)$ are



obtained by subtracting the tide data $h_{Tide}(t)$ from the water-level data $h(t)$. The average detided data $h_{Mean}(t)$ over the time interval $t_{mean}$ that starts from the most recent data $t_{now}$ are then obtained. If the absolute value of the $h_{Mean}(t_{now})$ exceeds the absolute value of the $h_{Mean}(t)$ with the bounds of the given threshold $TH_{TIDE}$, then an alarm(TIDE) is triggered.

Any kind of alarm that is triggered lasts for a time interval $t_{alarm}$. To minimize the possibility of a false alarm, we introduced the tsunami detection index (TDI), which is divided into five levels depending on the number of different types of alarms triggered simultaneously. If these alarms are triggered simultaneously, TDI increments by one.

# 3 Results

Fig. 7 shows nine examples of results obtained using the outlier detection algorithm based on calibrated thresholds. We calibrated 17 thresholds to identify the parameter configuration that ensures the best performance for the Ulleung-do surge gauge data. Table 2 shows the final values of thresholds used in this study. Fig. 7(a) shows a case in which two types of outliers are clearly detected: one that slowly increases and gradually decreases, and another that sharply increases and suddenly decreases. Fig. 7(b) and (c) show cases in which the long-term outliers that offset the data are also perfectly detected. Coincidentally, we found that the times of the starting and end points of the outliers are extremely similar, whereas the offset moves in the opposite direction. Fig. 7(d) and (h) present cases in which the long-term outliers with gaps are clearly detected using the keeping method, which allows the algorithm to manage the missing data. Fig. 7(e) shows the double outliers. Here, the second outliers appear immediately after the first outliers. Fig. 7(f) shows the meteorological outliers when a wind wave advisory was in effect for the far East Sea. Fig. 7(g) presents a case in which approximately 10 min of gaps exist in the data stream. It should be noted that the point immediately after gaps is classified as normal by the keeping method. Fig. 7(i) shows the instantaneously oscillating outliers that maintain for approximately 10 min. Because the performance of the gap-filling algorithm is in inverse proportion to the gap size, we should focus on the points classified as normal among outliers in Fig. 7(a), (e), and (i) that prevent long-term gaps. The results show that the outlier detection algorithm is unique in detecting outliers from the water-level data of the Ulleung-do surge gauge.

After calibrating the thresholds of the outlier detection algorithm, we calibrated five parameters of the gap-filling algorithm that ensure the best performance for the Ulleung-do surge gauge data. To calibrate the parameters, the water-levels were intentionally omitted and gaps were predicted. Furthermore, the performance tests were conducted with respect to the statistical parameter between the predicted water-levels and measured data concealed in the gap-filling process. Table 3 shows the final values of parameters used in this study. We used the same values for $mv_{LGFA}$, windowsize and npastdata, as in Lee and Park (2015). Because using either a large N_inter or a single point from the opposite ends of the gaps for linear interpolation can distort a slope provoked by temporary water-level fluctuation, two points were used for N_inter. Fig. 8 shows the mean of the correlation coefficient (r) and the root mean square error (RMSE) between the predicted and measured data when the values of $mv_{LGFA}$, windowsize, npastdata, and N_inter remain the same. Thirty-four cases of gap sizes from 3 to 36 h with 1-h intervals, and 31 datasets from March 12, 2011 13:55:10 (or 100,000 steps away from March 1,



2011 00:00:00) to March 29, 2011 22:33:40 (or 250,000 steps away from March 1, 2011 00:00:00) with 50,000 s intervals (or 5,000 steps) were considered. As the gap size increases from 3 to 36 h, the r of SGFA decreases from 0.79 to -0.01 and then slightly increases to 0.07, whereas the r of LGFA fluctuates in the range of 0.66 and 0.87, as shown in Fig. 8(a). In addition, the RMSE of SGFA increases from 1.32 to 8.68 cm and then slightly decreases to 8.01 cm, whereas the RMSE of

LGFA increases from 1.28 to 5.21 cm until the gap size is 12 h. Then it fluctuates in the range of 4.54 and 6.28 cm, as shown in Fig. 8(b). Because the SGFA is much faster than the LGFA, $n_{LGFA}$ was set to 4 h where both r and RMSE of SGFA are similar to those of LGFA.

Fig. 9 shows four examples of the results obtained with the LGFA and Table 4 provides the estimates of the RMSE, MAE, and r for those four examples. Fig. 9(a) and (b) show cases in which the water-levels of the target and search data are similar,

but the difference between the rest of the water-levels of the selected search window data and the measured data increases as time passes. However, because the EPFM enforces the end point of the selected search window to match the end point of the measured data, the SWEP data shows good agreement with the measured data. Fig. 9(c) and (d) show cases in which the difference between the SWEP and measured data remained the same even after applying the EPFM because the end point of the selected search window data is similar to that of the measured data. We should note that even though comparatively

lower accuracy is obtained as the gap size increases, because the longest gaps of the Ulleung-do surge gauge between March 1st and March 31st are approximately 6 h, the gap-filling algorithm performs reliably in alleviating the gap-filling problem in the Ulleung-do surge gauge.

Fig. 10 shows the results of the outlier detection and gap-filling algorithms when applied to the one-month data of the Ulleung-do surge gauge. Based on the results of a numerical simulation (Lee et al., 2015), the event period was set to 8 h

immediately following the 2011 Tohoku earthquake. We should note that the tsunami signal is preserved during the event period (see yellow box in Fig. 10(a)). Because several equivocal outliers are hardly distinguishable and no exact solution exists for real gaps, we compare the outlier-removed and gap-filled data with the predicted tide using T_TIDE, as shown in Fig. 10(b). Fig. 10(c) and (d) reveals that most of the outliers that stand out from the majority data are successfully removed. In addition, we find that the area of the scatter plot slightly decreases after applying the gap-filling algorithm. In other words,

the gap-filled data successfully follows the trend of the predicted tide.

Fig. 11 shows the performance of the tsunami detection algorithms. The left panels show the three-day segment of the time-series spanning from March 10, 2011 00:00:00 onwards to March 13, 2011 00:00:00, whereas the right panels zoom in on the time of the 2011 Tohoku earthquake. Even though the Ulleung-do surge gauge goes offline on March 10, 2011 at 03:10:00 and restarts the recording on March 10, 2011 at 09:29:00, because of the gap-filling algorithm, every tsunami

detection algorithm could be applied immediately following the recording restart. The background color in Fig. 11(a) indicates the TDI. When the TDI is equal to 4 or all alarms are triggered, the background color changes to red; when the TDI is equal to 3, the background color changes to orange; finally, when the TDI is equal to 2 or 1, the background color changes to yellow. Unlike other alarms, the control function CF of the SLOPE algorithm triggers the alarm only near the time of the 2011 Tohoku earthquake during the event period. Thus, the red alarm only appears on March 11, 2011 at 14:58:30 or



approximately 12 min after the 2011 Tohoku earthquake struck and remains for approximately 15 min. The yellow alarm with intermittent orange alarms is triggered outside the event period approximately 19 h after the 2011 Tohoku earthquake, where the abrupt descending water level was measured by an unknown reason.

## 4 Discussion

An aerial ultrasonic gauge was installed in the Ulleung-do rather than a float-driven gauge because a float-driven gauge generates slow and nonlinear responses to tsunamis (Joseph, 2011). However, the aerial ultrasonic gauge is exposed to high frequency oscillations induced by wind waves and meteorological events as shown in Fig. 2. To remove contaminated data without distorting a tsunami wave, we apply the concept of the event period in which the outlier detection and gap-filling algorithms stop, whereas the tsunami detection algorithm starts to run during the event period. We should note that the event period is a crucial factor that directly affects the performance of the TADS because a short event period might miss a true tsunami wave, whereas a long event period might confuse outliers with a tsunami wave, thus generating a false alarm. The results of our study show that the overall performance of TADS is reasonable for detecting a tsunami during the event period (Fig. 11). However, for further improvement, the start and end points of event periods should be more carefully examined.

The start point of the event period is initially designed to be based only on seismic information for detecting a tsunami because most tsunamis are caused by seaquakes. According to the tsunami database, seaquakes are responsible for approximately 82% of tsunamis (Joseph, 2011). However, tsunamis can result from submarine landslides, terrestrial landslides, volcanic eruptions, atmospheric disturbances, asteroid and comet impacts, and man-made explosions (Pugh and Woodworth, 2014). Korea experienced meteo-tsunamis in both 2007 and 2008, which engulfed a part of the western coast of Korean Peninsula, causing two and nine casualties, respectively (Yoon et al., 2014). Thus, in order to recognize several types of tsunamis, including seismically generated tsunamis, the TADS should link with information related to landslides, volcanic activity, and atmospheric pressure, etc.

The end point of the event period was set to 8 h after the 2011 Tohoku earthquake struck. Numerical simulation shows that 8 h is sufficient for tsunami to pass the Ulleung-do (Lee et al., 2015). However, it should be noted that the Japan Meteorological Agency (JMA) experienced a high tsunami wave caused by the 2006 Kuril Island earthquake after hasty cancellation of alarm (Kamigaichi, 2009). The unexpected high tsunami wave occurred because the domain of the numerical simulation (mesh) was set to a small area, limited around Japan could not adequately represent the reflected wave as a result of the unique topography of the Pacific Ocean. In addition, tsunami travel time delay is found in a numerical simulation of distant tsunamis. This delay is caused by elasticity and gravitational potential variation (Watada et al., 2014). Thus, in order to consider all possible tsunami waves, the event period of TADS should be defined based on a global tsunami prediction system that considers the travel time delay of distant tsunamis.

Not only the event period but also the four thresholds of the tsunami detection algorithm are directly related to the sensitivity and accuracy of TADS. In this study, the thresholds ($TH_{DART}$, $TH_{IS}$, $TH_{CF}$, $TH_{TIDE}$) were determined to be 5 cm, 0.01 cm/s, 4,



and 5 cm, respectively, based on the case of the 2011 Tohoku earthquake. We performed a sensitivity test based on these thresholds (Fig. 12) to facilitate discussion. After normalizing the calibrated thresholds to 1, we changed the thresholds from 0 to 2 with 0.02 intervals. In other words, $TH_{DART}$ and $TH_{TIDE}$ were set from 0 to 10 cm, whereas $TH_{IS}$ and $TH_{CF}$ were set from 0 to 0.02 cm/s and 0 to 8, respectively. Fig. 12(a) shows the time-series of the Ulleung-do surge gauge starting from the origin time of the 2011 Tohoku earthquake. A steep fluctuation in water level follows a small fluctuation in water level that begins approximately 8 min after the earthquake. The colored circle represents the start point of each TDI level when the normalized threshold is set to 1. Fig. 12(b) and (c) show that the alarm rate and start time of the alarm depend on the normalized thresholds. As the normalized thresholds increase, the alarm rate decreases and the alarm(CF) of the SLOPE algorithm stops triggering when the normalized threshold is greater than 1.34. Even though alarm(IS) and alarm(CF) derive from the same slope-based algorithms, the alarm(CF) yields more sensitive results than does alarm(IS). By contrast, two amplitude-based algorithms (DART and TIDE) show similar patterns of alarm rate and start time. Fig. 12(d) and (e) show that the TDI rate and start time depend on the normalized thresholds. As the normalized thresholds increase, the TDI rate of Level 4 decreases and disappears when the normalized threshold is greater than 1.34. Because the start time of alarm(IS) increases dramatically when the normalized threshold is greater than 1.74, the TDI rate of Level 3 decreases simultaneously. We should note that if the normalized threshold is set to less than 0.4 ($TH_{DART}$ = 2 cm, $TH_{IS}$ = 0.004 cm/s, $TH_{CF}$ = 1.6, $TH_{TIDE}$ = 2 cm), one or more alarms are triggered within 8 min, which thus represents a false alarm (Fig. 12(c)). By contrast, if the normalized threshold is set to greater than 1.06 ($TH_{DART}$ = 5.3 cm, $TH_{IS}$ = 0.0106 cm/s, $TH_{CF}$ = 4.24, $TH_{TIDE}$ = 5.3 cm), the start time in which the TDI equals 4 requires 16 min or more, which is the moment when the steep fluctuation in water level has already passed (Fig. 12(e)). Thus, to prevent both a false alarm and delay of alarm, the normalized threshold should be set in the range of 0.4 to 1.06. We then set the value as 1 ($TH_{DART}$ = 5 cm, $TH_{IS}$ = 0.01 cm/s, $TH_{CF}$ = 4, $TH_{TIDE}$ = 5 cm) to avoid false alarms as much as possible.

To calibrate the TADS for minimizing the false alarm rate and detection time, the TADS should first be tested based on an extensive number of tsunamis. However, because of the lack of tsunami records in the Ulleung-do surge gauge, the present study deals only with the case of the 2011 Tohoku tsunami. For stations having insufficient tsunami records, Beltrami and Risio (2011) tested their tsunami detection algorithm with synthetic tsunami signals in which ideal sinusoidal tsunamis were superimposed on ideal wind-waves based on Jonswap wave-spectra. In addition, Bressan et al. (2013) tested their tsunami detection algorithm with synthetic tsunami signals in which the results of numerical simulation were superimposed on the tide gauge record of possible circumstances (e.g., calm or rough sea). Risio and Beltrami (2014) tested their tsunami detection algorithm with historical tsunami signals, in which the record of the DART buoy was superimposed on the wind wave, which in turn was synthesized by means of the random-phase method. Thus, we plan to conduct more detailed tests of TADS using both synthetic and historical tsunamis.



## 5 Conclusion

We proposed a tsunami detection system called TADS to apply to discontinuous time-series data with outliers using the records of the Ulleung-do surge gauge. The TADS comprises three major algorithms designed to update at every new data acquisition. These algorithms are used for outlier detection, gap filling, and tsunami detection. To distinguish a tsunami from a record containing outliers and gaps, we proposed the concept of the event period. The performance of TADS was demonstrated using the record of the 2011 Tohoku tsunami. The results show that the overall performance of TADS is reasonable in detecting a small tsunami signal superimposed on both the outlier and gap. However, because the proposed thresholds and parameters do not guarantee that the TADS will detect all tsunamis measured in the Ulleung-do surge gauge to a satisfactory degree, sufficient tests of TADS are required. We want to stress that the efficiency and simplicity of the algorithms used in the TADS enable its wide application in tsunami monitoring areas. To enhance the tsunami monitoring network, an extension of the TADS to include the data of related organizations such as the records of 46 tide stations (using a 1-min sampling interval) operated by the Korea Ocean Observing and Forecasting System are currently being developed.

## Acknowledgments

This study was supported by the "Research and Development for KMA Weather, Climate and Earth System Services" project of the National Institute of Meteorological Sciences.

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





**Table 1.** Parameters of tsunami detection algorithm.

| Name | Value | Name | Value | Name | Value |
|---|---|---|---|---|---|
| $\omega_0$ | 2.1957 | $t_{GTide}$ | 16 min | $t_{mean}$ | 1 hour |
| $\omega_1$ | -2.2038 | $t_{Tide}$ | 1 hour | $t_{FP}$ | 2 days |
| $\omega_2$ | 1.3233 | $t_{sm}$ | 6 min | $t_{BP}$ | 10 days |
| $\omega_3$ | -0.3152 | $t_{BS}$ | 1 hour | $t_{sample}$ | 1 min |
| $TH_{DART}$ | 5 cm | $TH_{IS}$ | 0.01 cm/sec | $TH_{TIDE}$ | 5 cm |
| $t_{IS}$ | 10 min | $TH_{CF}$ | 4 | $t_{alarm}$ | 10 min |
| $t_G$ | 15 min | inter | 1 min | | |

**Table 2.** Parameters of outlier detection algorithm.

| Name | Value | Name | Value | Name | Value |
|---|---|---|---|---|---|
| THh | 7 cm | TH_4 | 3 cm | THS2 | 3 cm |
| TH4 | 3 cm | TH_3 | 9 cm | THS3 | 5 cm |
| TH3 | 3 cm | TH_2 | 10 cm | THS4 | 2 cm |
| TH2 | 3 cm | TH_1 | 10 cm | THD1 | 10 points |
| TH1 | 4 cm | TH_0 | 10 cm | THD2 | 20 points |
| TH0 | 4 cm | THS1 | 2 cm | | |





**Table 3.** Parameters of gap-filling algorithm.

| Name | Value | Name | Value | Name | Value |
|---|---|---|---|---|---|
| $mv_{LGFA}$ | 10 min | $n_{LGFA}$ | 4 hours | npastdata | 100 |
| N_inter | 2 points | windowsize | 2 | | |

**Table 4.** Starting time of target and search windows, and the performances of four cases.

| Gap Size (hr) | Target Window Starting Time | Search Window Starting Time | RMSE (cm) | MAE (cm) | r | MIN(MAE) (cm) |
|---|---|---|---|---|---|---|
| 3 | Mar. 31, 2011 05:31:10 | Mar. 20, 2011 07:26:20 | 0.72 | 0.58 | 0.96 | 0.39 |
| 12 | Mar. 21, 2011 21:07:10 | Mar. 19, 2011 07:55:10 | 2.61 | 2.08 | 0.96 | 1.73 |
| 24 | Mar. 17, 2011 15:21:40 | Mar. 03, 2011 16:19:10 | 3.85 | 3.22 | 0.94 | 2.78 |
| 36 | Mar. 21, 2011 15:36:00 | Mar. 18, 2011 01:26:20 | 5.55 | 4.56 | 0.59 | 2.97 |





**Figure 1: Location of the Ulleung-do surge gauge (130.913°E, 37.538°N).**




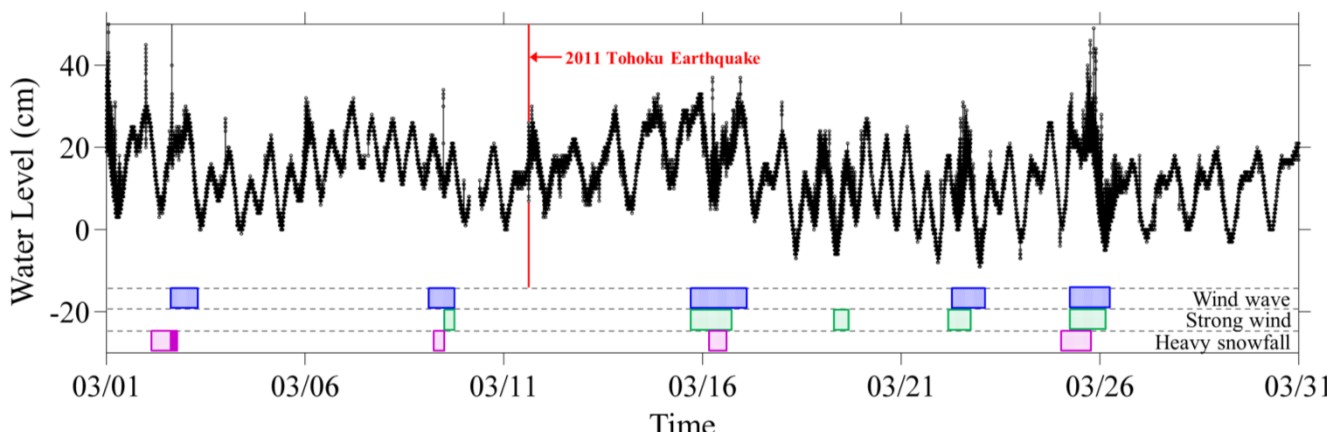

**Figure 2: Time-series of the Ulleung-do surge gauge with a special weather report from the Korea Meteorological Administration (KMA). The dotted rectangles represent the following advisories: wind wave (blue), strong wind (green), and heavy snow (purple). The filled rectangle represents a heavy snow warning (purple) and the red vertical line indicate the time of the 2011 Tohoku earthquake.**

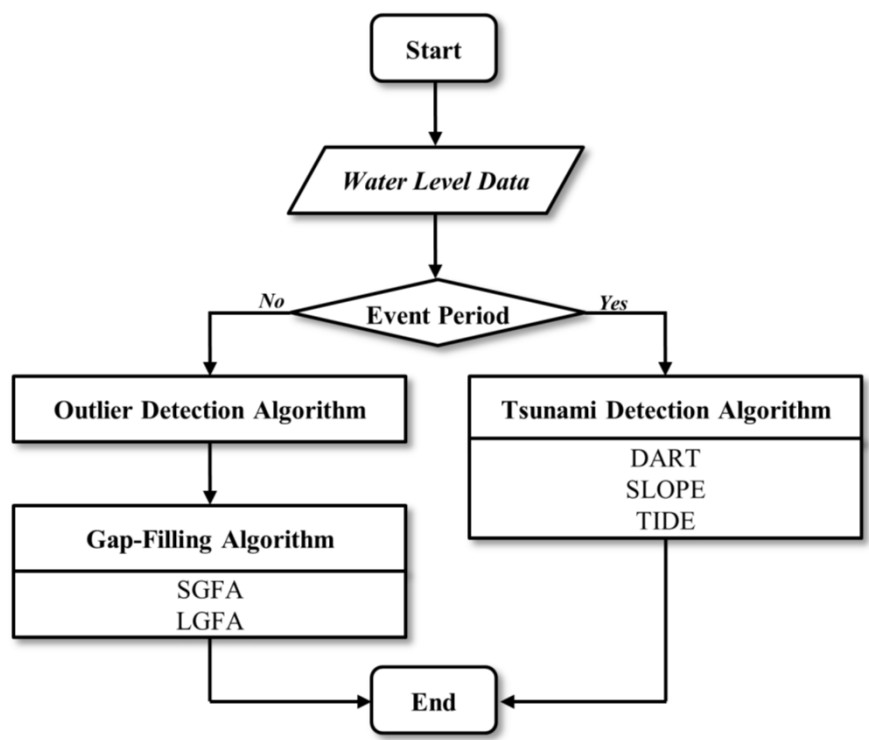

**Figure 3: Flow of the proposed tsunami arrival time detection system.**




**Figure 4: Flow of the outlier detection algorithm. Rounded rectangle nodes represent the terminal where the process starts and ends. Parallelogram and diamond nodes represent data and decision, respectively.**






**Figure 5: Flow of the gap-filling algorithm. The sequence of the colored rectangles represents the measured data, whereas the white rectangles represent the gaps.**




**Figure 6: Flow of the tsunami detection algorithm. Blue circles represent the past data used for each algorithm, and red circles represent the predicted data.**




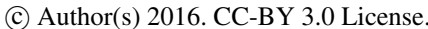

**Figure 7: Examples of the outlier detection algorithm. The gray line with a circle represents the original data, and the black line with a circle represents the outlier-removed data.**





**Figure 8: Calibration of the gap-filling algorithm: (a) correlation coefficient (r), and (b) root mean square error (RMSE). The black line with a circle represents the results of SGFA, whereas the red line with a circle represents the results of LGFA. Standard deviations for each gap size are indicated by the error bars.**





**Figure 9: Examples of the gap-filling algorithm. The left panel shows the time-series when the gap sizes are: (a) 3 h, (b) 12 h, (c) 24 h, and (d) 36 h. The black line represents the data of the target window, and the gray line represents the data of the selected search window. The blue line represents the predicted or SWEP data, and the red line represents measured data that was intentionally omitted. The right panel shows the scatter plot when the gap sizes are: (e) 3 h, (f) 12 h, (g) 24 h, and (h) 36 h. The colored point represents the frequency of data plotted inside the circle with a radius of 1 cm.**




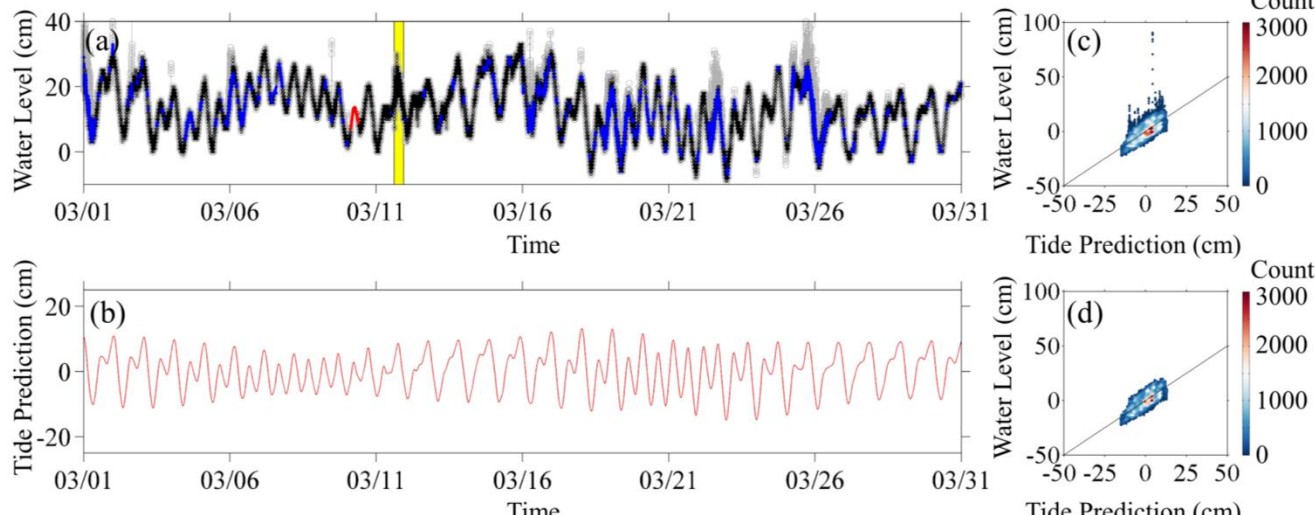

**Figure 10: Performance test of the outlier detection and gap-filling algorithms: (a) time-series of the Ulleung-do surge gauge. The gray line represents the original data, and the black line represents the outlier-removed data. The blue and red lines represent the predicted data using the SGFA and LGFA, respectively. The yellow box represents the event period from March 11 14:46:00 to March 11 22:46:00; (b) predicted tide using T_TIDE; (c) scatter plot of tide and original data; (d) scatter plot of tide and outlier-removed and gap-filled data. The mean value of each axis in the scatter plot is fixed to 0 by subtracting the mean value of each data from its data. The colored point represents the frequency of data plotted inside the circle with a radius of 1 cm.**





**Figure 11: Performance test of the tsunami detection algorithm: (a) time-series of the Ulleung-do surge gauge. The gray line represents the original data, and the black line represents the outlier-removed data. The blue and red lines represent the predicted data using the SGFA and LGFA, respectively. The red, orange, and yellow boxes represent the TDI; (b) time-series of the DART index of DART algorithm; (c) time-series of the IS of SLOPE algorithm; (d) time-series of the CF of SLOPE algorithm; (e) time-series of the $h_{Detide}$ of TIDE algorithm. The red lines represent a threshold. Gray vertical lines indicate the tsunami detection times, where the black line exceeds the red line. In all plots, the blue vertical lines show the event period from March 11 14:46:00 to March 11 22:46:00.**




**Figure 12: Sensitivity test of the tsunami detection algorithm: (a) time series of the Ulleung-do surge gauge. The colored circle represents the start point of each TDI. Illustration of the trends of: (b) alarm rate, (c) start time of alarm, (d) TDI rate, and (e) start time of TDI.**