# Peer review of "Tsunami arrival time detection system applicable to discontinuous time-series data with outliers"

_Natural Hazards and Earth System Sciences, 2016_

## Referee Comment (RC1) · Anonymous Referee #1 · 29 Jun 2016

Tsunami arrival time detection system applicable to discontinuous time-series data with outliers, by Jun-Whan Lee, Sun-Cheon Park, Duk Kee Lee, and Jong Ho Lee, Nat. Hazards Earth Syst. Sci. Discuss., doi:10.5194/nhess-2016-153, 2016.

This paper describes a method for operational tsunami detection for the Korean tide-gauge of Ulleung-do. As pointed out by the authors, tsunami detection (and in general tide-gauge data processing) in real-time implies first to solve the problem of the management of data gaps and outliers. A method for outliers detection and gap-filling is implemented, together with three different algorithms for tsunami detection. The procedure is then tested on a month of data, which includes outliers, gaps and the signal of the 11 March 2011 tsunami.

The manuscript addresses an interesting and very important topic related to oper-

ational tsunami detection and warnings, obtaining interesting results. However, the manuscript has some weaknesses and it needs to be improved in several aspects, especially in the description of the algorithms. In my opinion, the manuscript needs to be revised in order to be published in nhess. To help the authors improve the manuscript, I am writing some comments and suggestions.

———

Specific comments

The language is generally good, except some small mistakes. However, starting with paragraph 2.1, the explanations become very difficult. The outlier and the gap-filling algorithms are not clearly explained, several important information are missing (or postponed to the end) and therefore it is very difficult to read. The authors should make a further effort to better explain the algorithms in detail. The schemes provided are also too complex and do not help to make the explanation clearer (figures 4, 5 and 6). Some additional explanations come from the results and the discussion, however it would be better to anticipate them in order to understand what is being done. The acronyms and symbols are too much alike and it is difficult to distinguish them. It would be easier to give very different names to parameters according to the outlier, gap-filling or the three tsunami detection algorithms.

The explanations should also clarify the following issues and information:

Methods

* In the abstract, it is clearly stated that the authors would make use of the concept of the event period, which is however addressed only at the end, in the discussions, while the event time is already used in paragraph 2.2. Its explanation should be anticipated.

Outlier algorithm

* Is the short and long outlier detection algorithm the same? How is the difference in wave height between neighboring points computed? Is it the difference of wave height

of the time "now" with the previous datum? The 8-points window is used only to detect long-term outliers or also single outliers? * How does the outlier algorithm search for outliers? What does it mean that the algorithm is accelerated? Which is the stopping condition?

Gap filling algorithm

* The explanation of the algorithm lacks some important details, so that the algorithm method, especially LGFA, cannot be understood. * How long is n_LGFA? Please explicit it in the text. * What do you mean with target data? How are the target data first estimated? Or what are the target and search data? How are they used? Do you use the target data to look for data in the past that could fill the gap? It is not clear. * What is EPFM? What is the SWEP data? Some acronyms are not defined.

Tsunami detection algorithms

* The description of the SLOPE and TIDE algorithms are not very clear. In particular, the explanation and the reference to the algorithm TIDE is ambiguous, since it is a tool for harmonic analysis and tide prediction and not for tsunami detection. It should be explained more clearly how this algorithm is used to set up a tsunami detection. * How is it used the tsunami detection index? Why is it computed? It should be explained before the end of the results.

Results

* The results for outlier detection are here presented as a list of figure descriptions. It should be better to give a general explanation of the algorithm performance and use the figures as examples. * The outlier algorithm seems very strict: from the few data shown, I would not mark the data in figure 7(h) or 7(g) as outliers, or even the data in 7(d) or 7(e), especially if the same data patterns happen often in the time series, as it could seem in figure 7(f). To discriminate outliers, it would be better to inspect time series longer than a month, in order to safely tell if the data shown are outliers or not.

Where is the Ulleung-do tide-gauge located? Have you considered the possibility that some of the data shown are long period waves of about 5 cm height, which could be common in harbors or bays? * Regarding the gap filling algorithm, the explanation of how the performance tests were computed is missing (page 7, line 24-25). It would be interesting to see the performance of gaps shorter than 3 h.

Discussions

* Some explanations should be anticipated, some other information are repeated from the introduction. In particular the explanations on how the event time works should be anticipated. * I understand that so far only earthquake generated tsunamis could be detected, since the event time is triggered only by earthquake information and, without the activation of the event mode, there is no tsunami detection. Is this correct? This requires further discussions. * There is some confusion about travel times and travel time delays. The sentences at page 9 lines 27-28 is not correct ("tsunami travel time delay is found ... This delay is caused by ... Watada et al, 2014"). Did you mean to take into account the propagation time of tsunamis? * How is the alarm rate and the TDI rate defined in the sensitivity test?

Additional remarks

* Tsunamis can be generated by earthquakes. The term earhquakes should be pre-ferred to seaquakes. Also terrestrial landslides is not correct: landslides that start over the land and fall in the water to generate tsunamis are usually called subaerial landslides. * Page 3, line 6: The descriptions of the literature algorithms need to be more rigorous, for example please check the description of the algorithm by Beltrami and Risio 2011. * Table 3 is never mentioned in the text. * Figure descriptions could be better explained. * In the introduction, a broader view of the situation of tsunami detection and automatic data-processing could be addressed, with the more recent developments. For example, new technologies have been introduced, and worldwide and European tsunami warning systems are being developed, together with automatic

data-processing algorithms.

---

## Referee Comment (RC2) · Anonymous Referee #2 · 30 Jun 2016

General comments: A novel tsunami arrival time detection system is proposed in the manuscript. The system consists of three separate modules: outlier detection algorithm, gap filling algorithm, and tsunami detection algorithm. System is tested on sea level time series measured at Ulleung-do (an island off shore of Korea) tide gauge instrument during March 2011.

Described tsunami detection method is interesting and worthy of further development and testing. However, before manuscript can be published I have two major comments: (1) Detection system is calibrated on too short (and too few) time series, namely only one month of sea level data including only one tsunami event (regardless of strength of this event), and this is simply not enough to test or calibrate such a system. Authors are aware of this short coming and say that it might be overcome by either using longer measured time series which include more tsunami events or by using synthesized time

series. I believe authors should follow their own suggestion. Certainly, there are a number of tide gauge stations in the World which have longer records and have more tsunami events in them, perhaps even in a nearby Japan. Or if authors are not able to obtain this data, then they should do their tests on synthesized time series, following some of the papers they quote.

(2) Manuscript is not very clearly written. With all the abbreviations, flow charts, tables without explanations , it is hard to understand some of suggested algorithms. Authors should make their text more clear. More specific comments on this will follow.

Specific comments: A number of terms are not explained when first introduced but only afterwards (and some never), e.g. event period is mentioned in the abstract, and several times after it, but not explained until results section.

In introduction what are: short-time outliers, long-term outliers, short gaps, long gaps? I understood after reading the manuscript that these values (length of time steps) are later defined through calibration for specific herein described tsunami detection system. However, in Introduction, when dealing with values from literature, you should write a range of these values used by other authors. It would also be good to say that for this particular tsunami detection system, values will be determined through calibration.

Also in Introduction, what are soft computing techniques?

Still in Introduction, there are two contradictory statements: (1) "A low probability exists for tsunamis to occur in the East Sea" (2) "This can be used to detect weak tsunami signals that are common in the Ulleung-do surge data" What is correct then? If there are more tsunami signals in the Ulleung-do surge data, why not incorporate longer time series with these signals into your analysis?

Figure 1. I suggest adding bathymetry contours (perhaps coloring the figure?)  and also pointing to Yamato rise mentioned in introduction.

Figure 2. Resolution should be increased. Why are sea levels showed with dots? I

think it's better to just use line. Also, it would be nice to add a zoomed in window showing Tokohu Earthquake period.

In methods, you again refer to long and short gaps without defining them or saying that they would be defined later.

In general, idea behind your process should be more clearly presented. Why do you remove outliers and fill in gaps when these algorithms are not used during the event? I assume to be able to compare event period time series to time series from previous time steps - but then this should be clearly stated. Also, what exactly do you do with outliers, remove them and then fill the gaps? I believe so, but this is not clearly written.

In 2.1. Outlier detection algorithm, entire chapter is highly difficult to follow. Try to simplify while still keeping the most important points. Likewise, Figure 4 is also very difficult to comprehend. I understand that it is a code flow chart - but perhaps here you could put a more simple version, and put this one (alongside with a code) to supplementary material? I.e. if your code is not described in some other paper. If yes, I do not see a need for a complicated figure and code.

In 2.2. Gap-filling algorithm, I have similar comments as for 2.1., although it is written a bit more clearly, and figure is more understandable and helps follow text (but still not good enough!). There are also a number of abbreviations - so it is easy to get lost. Some of this abbreviations are, I believe, not explained: what is SW, what is EPFM, what is SWEP? what is h1, h2, h3, h4, t1, t2.

As I understand, you basically copy search data to gap window - but before that you make sure that you fit first and last point of search data to first-1 and last+1 point of the gap? If so, this can be clearly written.

In sentence "A predefined length of points (N_inter) from the last poing... to create the linear interpolation..." does this mean that you linearly interpolate data by using the least square method? Or something else?

In 2.3. Tsunami detection algorithm you present Table 1. This table is completely unclear. I guess that first two columns are related to DART, second two to SLOPE, and last two to TIDE algorithm. This is nowhere written (should be in the table caption). Some of abbreviations in Table 1. are defined in text but most of them not. What is tIS, tG, tGTide, tTide, and so on?... I assume some parameters related to DART, SLOPE and TIDE equations. But if so, these equations should be written and explained in the manuscript. Also, are all of these calibrated values or general values related to method or a mix?

How is Tsunami Detection Index divided into five levels if you have only three tsunami detection algorithms? What are these five levels?

Figure 6. is also really difficult to follow. I'd say if all of algorithm you use (including outlier and gap filling algorithm) are from previous papers, there is no need in including such a complicated version of Figures 4, 5 and 6. Something simpler would do, or even omitting figures.

Related to Tables, none of them are very clear or fully explained. In Table 3, what does it mean that windowsize is 2 (two of what? Points, hours?...), or that npastdata is 100 (100 of what?).

In Table 4, why is search window located 3-14 days before actual gap?

In Results, you say that yellow alarm is triggered outside of the event period. How is this alarm triggered if you are not in the event period? And thus (from Figure 1) no tsunami detection algorithm should be activated?

In Discussion, I believe your method which is triggered only when there is an event, would be extremely difficult to use during events which have not-easily detectable sources (like meteotsunamis, landslide, ...). You can elaborate further.

Technical corrections:

Page 5. line 22. "start and end points" instead of "end points".

---

## Author Comment (AC1) · 14 Oct 2016

Dear Reviewer:

Please find the attached file which contains the revised manuscript and the detailed response to your comments.

Best regards,

Jun-Whan Lee

Please also note the supplement to this comment:
http://www.nat-hazards-earth-syst-sci-discuss.net/nhess-2016-153/nhess-2016-153-AC1-supplement.zip